# Infection, Allergy, and Inflammation: The Role of *Aspergillus fumigatus* in Cystic Fibrosis

**DOI:** 10.3390/microorganisms11082013

**Published:** 2023-08-05

**Authors:** T. Spencer Poore, Edith T. Zemanick

**Affiliations:** 1Department of Pediatrics, University of Alabama at Birmingham, Birmingham, AL 35223, USA; 2UAB Gregory Fleming James Cystic Fibrosis Research Center, Birmingham, AL 35223, USA; 3Department of Pediatrics, University of Colorado Anschutz Medical Campus, Aurora, CO 80045, USA; edith.zemanick@childrenscolorado.org; 4Breathing Institute, Children’s Hospital Colorado, Aurora, CO 80045, USA

**Keywords:** *Aspergillus*, fungi, ABPA, airway infection, culture, host response

## Abstract

*Aspergillus fumigatus* (Af) is a mold frequently detected in airway samples from people with cystic fibrosis (pwCF). Abnormal airway mucus may allow Af to germinate, resulting in airway infection or an allergic response. While Af is known to increase morbidity in pwCF, individual responses and the degree of impact on lung disease vary. Improved approaches to diagnosis, treatment, and prevention of Af, particularly the persistent Af infection, are needed. This update highlights our current understanding of Af pathophysiology in the CF airway, the effects of Af on pwCF, and areas of research needed to improve clinical outcomes.

## 1. Introduction

Cystic fibrosis (CF) is one of the most common lethal genetic conditions affecting people worldwide [1,2]. It is characterized by a dysfunction of the cystic fibrosis transmembrane receptor (CFTR) protein that results in impaired chloride movement across epithelial surfaces [3]. These affected epithelial surfaces exist throughout the body, but the dysfunction in the gastrointestinal and pulmonary systems causes many of the most serious clinical consequences for pwCF [4]. Within the airways, the CFTR dysfunction impacts ion movement across the cell membrane surface into the airway lumen. This in turn impairs the flow of water across the membrane and thus mucus hydration within the airways. This phenomenon creates inspissated, dehydrated mucus within the airway lumen and impairs mucociliary clearance [5,6,7,8,9]. Given this, cilia beat is impaired, white blood cell mobilization is limited within the mucus, and both host and foreign matter becomes lodged in the abnormal mucus [10,11]. From this obstructive process, a cycle of inflammation and chronic infection begins with lack of mobilization of trapped bacteria and particulate. This results in airway destruction, increased mucus burden, and progressive obstructive lung disease [12,13,14]. While bacteria are the main culprits of infection and inflammatory response in the CF airway, many molds are implicated as also causing disease either by an infectious process or an allergic response. *Aspergillus fumigatus* (Af) is one of the most common molds isolated in airway samples from people with CF (pwCF) [15,16,17,18]. Despite its prevalence, few guidelines on the treatment of Af in pwCF exist and there is limited mechanistic understanding of why and how Af persists in the CF airway. Research is needed to understand the role of the Af disease in pwCF and, in particular, how new CF therapies, CFTR modulators, impact the frequency and pathogenicity of the Af infection.

## 2. Pathophysiology of Af in CF

*Aspergillus fumigatus* is a filamentous environmental mold that is transmitted via aerosolized spores [19,20]. These spores germinate into hyphae, allowing for active nutrient ingestion, growth, and eventual reproduction. The organisms release catalytic enzymes to digest nutrients in its environment and utilize multiple energy sources for growth and development [21]. *Aspergillus fumigatus* is often found in areas that are wet and warm in character, rich in soil and decaying organic matter, and with limited air circulation, i.e., damp basements, closets, or air conditioning units [22].

Environmental exposure to Af occurs frequently with spores inhaled into the airways. In the typical host, the spores of Af are trapped by mucus within the bronchi. Here, the spores are either ingested by macrophages and neutrophils or expelled through a mucociliary escalator made of mucus entrapment with subsequent ciliary expulsion out of the airway. This results in spore removal without persistent germination [20,23]. In pwCF that inhale these spores, mucus abnormalities, impairment in the mucociliary escalator, and altered immune response may trap conidia [1,24,25,26,27]. This allows the Af conidia germinating in a warm and nutrient rich environment. The spores develop hyphae and begin to produce enzymes, virulence factors, and antigenic epitopes [28]. As germination expands, hyphae can extend in multiple directions. In some hosts, it is suspected that Af anchors to the epithelium, invades the space between cell junctions, and can even become angioinvasive depending on the immune system of the host [29]. It is here that infection and inflammation triggering is thought to begin, recruiting macrophages, neutrophils, and eosinophils to the airway mucosa. This creates a vicious cycle of increased abnormal mucus production, cell death, and nutrient availability for Af [19,20,28].

While infection is likely playing a role in Af in pwCF, the allergic response to the organism contributes to airway damage and progression of disease in others. Af has been shown to release proteases that causes a type 1 hypersensitivity (IgE) response in some pwCF which in turn leads to eosinophilic recruitment, histamine release, and other Th2 processes within the airway. While the exposure is presumed to be in the airway, it is unclear whether the germination of Af in the mucus, at the level of the epithelium/cilia layer, or at the invasion of the epithelial junctions is what results in this pathway and triggering. Given the complexity of Af growth and impaired host responses in CF, little is known mechanistically of where “exposure” to Af actually occurs within the airway levels to induce these specific inflammatory responses.

## 3. Epidemiology of Af in CF

*Aspergillus fumigatus* is commonly detected in lower airway samples in people with CF with estimated prevalence ranging widely from ~10% to ~40%, with higher prevalence in studies using fungal-specific culture methods. In a study of 469 people with CF from centers across Europe and Australia, Af was detected from 16% of sputa from children and 38% from adults using a standardized fungal detection protocol [30]. Similarly, Hong and colleagues detected *Aspergillus* sp. (93% Af) and *Scedosporium* sp. from 41% and 5% of 487 sputum samples from adults (n = 211) using fungal specific culture media compared to the nationally reported prevalence numbers in the U.S. of 20.4% and 1.9% [31]. In a report from Germany over a five-year period, Af was detected in 29% of samples from 25,975 respiratory specimens including throat swabs, sputum and BAL samples, analyzed in a centralized reference laboratory (an average of 5195 samples from 637 patients per year). No change in *Aspergillus* sp. detection rate was seen over the 5 years [32].

Allergic bronchopulmonary aspergillosis (ABPA) is a pulmonary disease characterized by an allergic type 1 hypersensitivity, Ig-E-mediated response to *Aspergillus* sp. antigens manifesting in bronchiectatic change to the airway. This results in mucus plugging, increased sputum production, airway obstruction, and an asthma phenotype in these individuals [19,20]. In pwCF, the pathophysiology is thought to be related to the intrinsic CFTR defect resulting in inspissated mucus, trapping the allergens and allowing sensitization and subsequent T-helper 2 (Th2) reaction to the antigens [16,33,34,35]. A large meta-analysis found that the prevalence of Aspergillus sensitization and ABPA in pwCF was 39.1% (95% CI 33.3–45.1) and 8.9% (CI: 7.4–10.7) respectively, but noted a large amount of heterogeneity in criteria used to define these conditions between studies [36]. Furthermore, the prevalence of ABPA was higher in adults than in children. Af is the most common cause of ABPA, although other *Aspergillus* species including *A. flavus* and *A. niger* have been implicated. Non-aspergillus fungi (e.g., *Bipolaris* spp., *Penicillium* spp.) can result in allergic bronchopulmonary mycosis but are less common [37].

## 4. Role of Mucus in Af Acquisition in PwCF

There are many reports of altered biophysical and biochemical characteristics of mucus from pwCF compared to those without CF. Mucus is a complex network of hydrated and glycosylated compounds. These compounds are mucins that are long polymers of varying consistency and viscosity that are released by goblet cells within the airway mucosa [38,39]. In pwCF, given the mucosal CFTR dysfunction, these mucin networks become dehydrated and have altered biophysical properties at the epithelial surface. CFTR dysfunction with impaired chloride and bicarbonate transport also results in the dehydration of the airway surface liquid layer impairing the cilia movement. These changes restrict mucociliary clearance and allow persistence of antigenic agents [40,41,42]. There are many conflicting studies showing varying mucin expression, concentrations, production, and properties in pwCF [6,8,39,42,43,44]. Furthermore, mucin has been shown to change during pulmonary exacerbations in pwCF, showing increased expression of MUC5AC as well as altered pH in the CF mucus itself [5,7].

Limited studies have investigated the role of Af constituents on MUC5AC/B expression. Studies have shown that Af exposure can increase MUCAC expression in airway goblet cells [45]. Furthermore, IL-4, a Th2- or allergic-driven cytokine, also increase MUC5AC expression within the airway, further suggesting abnormal mucus secretion during an Af atopy interaction [46]. CF mucus itself has been found to have abnormal pH, altered mucin concentrations, biofilm promotion, and notably different mucociliary clearance than mucus from people without CF [5,7,8,41,43,47,48,49,50]. Studies have also shown that Af proteins signal an increase in MUC5AC secretion, potentially altering the mucus environment upon germination [45].

There is concern that other obstructive pulmonary conditions with abnormal mucus character or production may harbor Af disease similar to that of CF. Mucus from people with asthma, a condition associated with ABPA and Af allergy, has been shown to have excess mucus production and abnormal pH, showing an increasingly acidic mucus character as the severity of asthma increases [51]. Chronic obstructive pulmonary disease (COPD) is another pulmonary condition that has been shown to have increased mucus production and abnormal character, and it too has been shown to be associated with Af colonization and possible disease [26,52,53,54,55]. From this, further work regarding the biophysical characteristics, chemical makeup, and properties of mucus and Af germination is needed to understand the mechanistic actions of this organism.

## 5. Clinical Impact of Af in PwCF

Debate about the impact of Af on lung disease primarily hinges on whether the relationship between Af and worse disease are associative or causal [30]. Frequent co-infection with *Pseudomonas aeruginosa* and possible microbial community pressures resulting in selection of fungi by antibiotics and corticosteroids have made this challenging to unravel. In addition, Af causes different host responses, with some individuals developing an allergic, Th-2-type response resulting in ABPA or Af sensitization, while others develop a neutrophilic response more typical of CF endobronchitis. While ABPA has a defined clinical presentation with standardized criteria for diagnosis and treatment, the impact of Af in a non-ABPA phenotype is less clear [18]. In addition, some of the criteria for ABPA diagnoses overlap with progression of the CF lung disease (e.g., bronchiectasis) leading to diagnostic uncertainty in some patients.

Results from several clinical studies suggest that Af has detrimental effects on lung disease, even in cases without ABPA. Using data from the Australian Respiratory Early Surveillance Team (AREST-CF) program, Breuer and colleagues detected *Aspergillus* species in 11% of BAL fluid samples collected from children annually starting in infancy up to 6 years of age (n = 380 children; 1759 BAL samples) [17]. *Aspergillus* sp. was detected more often than *P. aeruginosa* (8%), and while *P. aeruginosa* and *S. aureus* decreased with progressive birth cohorts, *Aspergillus* sp. prevalence did not change. In multivariate analysis, age, pancreatic insufficiency, and inhaled tobramycin use were associated with the risk of the *Aspergillus* sp. infection. Annual chest computed tomography (CT) imaging was available for 330 children of whom 11% had *Aspergillus* sp. (75% of species were Af) [17]. Children with *Aspergillus* sp. had worse structural lung disease on CT and more progression of lung disease in the year after culture positivity. Changes on CT associated with *Aspergillus* most strongly were air trapping and mucus plugging. *Pseudomonas aeruginosa* was also associated with structural injury, but more strongly with bronchiectasis. Notably, only 4/115 children with the *Aspergillus* infection had total and *Aspergillus*–specific IgE levels consistent with ABPA. Airway inflammatory markers (NE, IL-8, and % neutrophils) were increased in those with *Aspergillus* sp., with the degree of inflammatory response similar to that seen with *P. aeruginosa*. Children also had worse clinical outcomes with more frequent intravenous antibiotic courses in the year after culture positivity and more cough and wheeze reported at the time of bronchoscopy. This longitudinal data from young children support a causative role for Af in lung disease progression rather than as a co-traveler with minimal impact outside of the setting of ABPA.

Hong and colleagues performed a cross-sectional study in adolescents and adults with CF aged 14 years and older to determine the association between Af detection from sputum and clinical status. Using fungal selective media, they detected Af in 10.3% of participants (n = 206) [56]. Quality of life, measured by the CFQ-R, a validated patient reported outcome measure in CF, was lower in those with Af compared to those without Af detected, which persisted when those with a diagnosis of ABPA were removed from the analysis. Other clinical characteristics, lung function (forced expiratory volume in one second, FEV_1_), body–mass index (BMI), *Pseudomonas* infection status, and pulmonary exacerbation status at the time of study did not differ between Af-positive and Af-negative groups. As in other studies, inhaled corticosteroid use was more frequent in the Af group; however, antibiotic use did not differ between groups [56]. Conversely, in a prospective study of 202 adolescents and adults with CF, O’Dea and colleagues found that positive Af culture was associated with lower lung function and increased frequency of pulmonary exacerbations, although in this study, Af positivity was not associated with quality-of-life measures [57].

In a retrospective, single-center study, our group compared clinical characteristics and outcomes in people with CF with frequent fungal positive cultures (most commonly Af) to those with no or rare fungi [58]. Individuals with frequent fungal positive cultures were more likely to have *Stenotrophomonas maltophilia* co-infection and *Pseudomonas aeruginosa* co-infection but were not more likely to have ABPA or lower lung function (culture positivity for Af is not a required criteria for a diagnosis of ABPA). When comparing those with rare fungal infection to those with frequent infection, there was no difference detected in lung function decline. Those with both ABPA and frequent fungal infection had a more clinically (but not statistically) significant decline in lung functions as well as a higher incidence of CF-related diabetes (CFRD) in comparison to those without ABPA and compared to those with ABPA but rare fungal positive cultures [58].

## 6. Role of Inflammation and Af in PwCF

Inflammation is a characteristic feature of CF airways and gastrointestinal disease, evidenced in elevated organ and systemic biomarkers of the host inflammatory response. In the airways, pwCFs typically have elevated cytokines indicative of neutrophilic inflammation. This is likely in response to distress and recruitment signals from the epithelium and resident macrophages in attempts to attack the bacteria within the airway. CFTR is also present in neutrophils, lymphocytes, and macrophages. The dysfunction of CFTR likely directly impairs host inflammatory response through abnormalities in chlorination, altered inflammatory pathways, and impaired pathogen killing [25]. This leads to a vicious cycle of bacterial growth in the abnormal mucus, increased cell debris and contents from increased neutrophil activity, utilization of cell materials by the bacteria, proliferation of bacteria, and then further recruitment of neutrophils [1,13]. CFTR-deficient T-cells and B-cells also may favor a Th-2 and exaggerated IgE response, predisposing individuals to ABPA [25].

Studies have shown increased IL-8, neutrophil elastase, and neutrophilic cell counts within bronchoalveolar lavage fluid from pwCF containing Af, suggesting a neutrophil response similar to that in response to bacterial infection [16,19,25,28,59]. *Aspergillus fumigatus* is also known to induce an allergic component in many individuals with atopy, and spore inhalation is the primary route of exposure as discussed in detail above. Some investigations have suggested abnormal eosinophil/allergy-driven response in CF, for example, showing that CF knockout mice have an increased IgE response, increased MCP-1 production, and increased eosinophil and neutrophil levels [60]. How these pathways interact within the airways of pwCF needs further investigation.

Multiple cytokines promote mucus secretion within the airway. These cytokines may alter the mucin concentrations secreted, the amount of mucus secreted, or other aspects depending on the disease processes underway. IL-4 and IL-5, key cytokines that are characteristic of the allergic Th2 immune response, may respond to Af exposure and alter mucin secretion within the airway, yet data are sparse and need further investigation [61,62]. IL-4 has specifically been shown to increase MUC5AC gene expression (similar to Af as described above), and further studies are needed in primary epithelial models [44]. IL-13, another canonical Th2 cytokine, has also been shown to increase mucus production via goblet cell hyperplasia, signaling a cellular change to the airway epithelium driven by eosinophils [63]. Interestingly, cytokines IL8 and IL1B, typical Th1-associated cytokines, have also been shown to regulate and stabilize mucus, indicating a neutrophil-derived pathway as well [40]. The differences in immune stimulation causing secretion of mucus may result in disparate mucus characteristics that may specifically promote Af germination. Augmented mucus stimulation induced by Af may also promote a feed forward loop that further promotes infection, yet further investigation is needed to understand this pathophysiology in pwCF.

## 7. Clinical management of Af Infections in PwCF

### 7.1. General Approach

The reliability of Af detection from airway samples varies depending on sampling type, specimen handling, and microbiologic culture conditions. Detection methods rely mostly on fungal culturing of an airway specimen for best organism growth and yield. Lower airway samples, sputum and bronchoalveolar lavage (BAL) are the preferred sample types. Oropharyngeal swabs, routinely used in the clinical setting for bacterial pathogen surveillance in non-expectorating patients, is less reflective of the lower airway and is not recommended for routine fungal surveillance. Needle core biopsies, or tissue resection, are generally reserved for more invasive or progressed fungal disease rarely seen in pwCF outside of organ transplantation requiring immunosuppression, although invasive aspergillosis pneumonia and aspergillomas (fungal balls) have also been reported in immunocompetent individuals with ABPA [20,21].

Approach to Af infection differs among CF care providers. Hong and colleagues surveyed CF Centers in the US, Canada, and the European Union. They found substantial variability in the diagnosis and treatment of Af infection. Regional differences in frequency of fungal culture screening were clear with EU countries more often testing at every clinic visit compared to centers in the US and Canada. Approaches to Af bronchitis and ABPA differed between North America and the EU. Despite the published criteria for ABPA diagnosis and management, there were differences in both the availability of testing and in the use of diagnostic markers to define ABPA [64].

### 7.2. ABPA

Clinical criteria for the diagnosis of ABPA in CF were published in a CF Foundation Consensus document and the American Academy of Allergy, Asthma, and Immunology, among others [65,66,67]. While there are variations between criteria, all include clinical deterioration, biochemical markers of hypersensitivity to Af (elevated total and specific-Af IgE, immediate cutaneous response to skin prick test, serum IgG or precipitating antibodies to Af) and evidence of structural disease consistent with ABPA (e.g., chest radiograph with infiltrates, mucus plugging or bronchiectasis). Serum total IgE levels are often >1000 IU/mL, but values >500 IU/mL should lead to additional testing. The mainstay of treatment for ABPA are systemic corticosteroids and/or antifungal therapy, although the risk of adverse effects from long-term steroids are high and many antifungal medications have a complex side effect profile. A full discussion of treatment of ABPA is beyond the scope of this review and we refer the reader to several recently published comprehensive reviews of ABPA [67,68]

### 7.3. Fungal Bronchitis

A phenotype of the Af infection called fungal bronchitis has been described. Patients with this diagnosis may have recurrent positive fungal cultures and evidence of worsened disease (lung function decline, changes in CT findings), poor response to antibiotic treatment of exacerbation, exacerbation symptoms not attributable to other causes, or elevation in Af-specific IgG, but do not have evidence of a hypersensitivity response [18,23]. Treatment is often initiated with antifungals, although data around their efficacy is limited. One single-center study attempted to evaluate the use of itraconazole using a randomized trial design. Aaron and colleagues performed a randomized controlled pilot study of itraconazole for patients with CF and positive Af cultures. Over 24 weeks, they did not detect a difference in frequency of exacerbation, lung function decline, or in patient-reported quality-of-life outcome measures [69]. Using retrospective registry data from Sweden, the investigators examined the impact of antifungal therapy in 42 patients with Af-positive cultures (19 treated with antifungals). The treated group had a more pronounced decline in lung function during follow-up compared to the untreated group, suggesting no benefit from antifungal therapy in the setting of the positive culture without symptoms, but likely reflecting a selection bias for worse disease in those receiving treatment [70].

### 7.4. Other Types of Aspergillus Disease

It is well understood that there are a variety of Af presentations and disease types in pwCF [23]. Hong and colleagues, and several study groups working in the UK, surveyed providers on the care of pwCF and fungal disease and found notable variability in diagnostic work-up, definitions of fungal disease in CF, treatment practices, and the understanding of fungal disease in CF [64,71,72,73]. For example, clinical CF centers vary in how often they perform fungal cultures and when they elect to initiate antifungal therapies. Differences in practice patterns and the lack of multicenter randomized clinical studies limit evidence-based clinical care guideline development.

There are studies showing an increase in allergic response to Af in some pwCF who do not meet diagnostic criteria for ABPA [74]. While many of these investigations were conducted some time ago on earlier birth cohorts, pwCF were generally found to have more frequent IgE-positive allergy testing to various allergens including that of Af [75,76,77]. This was seen to increase with age and associated with symptoms consistent with asthma and other atopic conditions [36,78]. Why this occurs can only be speculated, as little mechanistic study on the allergic response in pwCF has been continued into the modern era. Furthermore, the prevalence compared to other populations and pulmonary conditions is not well understood and needs further epidemiologic characterization. It is not uncommon for pwCF to have asthma- or atopic-like symptoms, and similar inhaled atopic therapies used in asthma are also used by pwCF in their routine care every day. Other conditions including invasive pulmonary aspergillosis and chronic necrotizing aspergillosis, caused by invasion of lung parenchymal tissue by *Aspergillus,* are most often found in individuals with severe immunocompromise and are rare in pwCF. Individuals with underlying chronic lung disease and airway structural injury (e.g., bronchiectasis, cavitary lesions) including chronic obstructive pulmonary disease (COPD) and CF are at risk for aspergillomas, fungal balls which develop in pre-existing cavities, although these are also rare in CF. Antifungals are the primary treatment for invasive pulmonary aspergillosis. In the case of aspergillomas, surgical resection is often required [20].

## 8. Relationship between Airway Bacterial Pathogens and Af in CF

Co-infection with specific bacterial infections and the presence of other clinical characteristics are associated with the Af infection in pwCF. Multiple studies have found associations with *P. aeruginosa* and *S. maltophilia* and Af acquisition, showing that in many instances, pwCF isolate these Gram-negative organisms prior to Af detection [58,79,80,81]. In particular, *P. aeruginosa* has been implicated as having some role in Af infection, yet the exact mechanism is unknown. Studies have shown conflicting reports of *P. aeruginosa* inhibiting Af growth in vitro as well as changes in biofilm activity between the two organisms [82]. These exact mechanisms for these findings are unclear. Future polymicrobial infection research is needed to better understand how these organisms interact, infect, and compete with one another in the CF airway [81].

Associations have also been found between treatment with intravenous and inhaled antibiotics (e.g., tobramycin) and chronic azithromycin and subsequent Af infection [83]. This suggests a shift in airway microbiota, alteration of the airway by Gram-negative organisms, or an inflammatory driven promotion of Af growth [58]. As more and more antibiotic regimens develop and change, further research is needed on the effects of microbial diversity and Af persistence within the airway in these polymicrobial environments.

## 9. CFTR Modulators and Aspergillus

Over the past decade, small-molecule therapeutics called CFTR modulators have been developed that increase functioning of the CFTR protein that is reduced or dysfunctional in pwCF and certain genetic variants. Most recently, elexacaftor/tezacaftor/ivacaftor (ETI) was FDA approved for people with CF and at least one F508del-CFTR variant (the most common variant) or another variant shown to be responsive to ETI in vitro. In April 2023, the FDA extended this approval down to 2 years of age, making ETI available for ~90% of the CF population in the US [84]. In clinical trials and post-approval longitudinal studies, ETI has been shown to substantially improve lung function, dramatically reduce pulmonary exacerbations, improve quality-of-life measures, and improve nutritional status [84].

Longitudinal data from the CFTR modulator ivacaftor, which has been approved for a small population of pwCF for over 10 years, is informative. Heltshe and colleagues found that the odds of culture positivity with *Aspergillus* sp. After one year was reduced (OR, 0.47, *p* < 0.05), as were the odds of *P. aeruginosa* positivity (OR 0.65, *p* < 0.001) compared to the year prior to treatment [85]. Similarly, retrospective data from the UK CF Registry found a modest reduction in prevalence of *Aspergillus* sp. in people on ivacaftor over up to 4 years. Whether similar or better long-term results are seen with ETI is yet to be seen [86].

One consequence of ETI therapy is a reduction in mucus burden resulting in less spontaneously expectorated sputum, requiring greater reliance on throat swabs for bacterial surveillance [87,88]. While the prevalence of bacterial pathogens may decrease, as was seen in long-term studies of ivacaftor in people with an ivacaftor-responsive mutation, infection is likely to remain a problem in people with CF, and new ways of monitoring for lower airway infection are needed. Recently, results from a longitudinal study of sputum collected from people receiving ETI for 12 years and longer (PROMISE, NCT04038047, www.clinicaltrials.gov accessed on 22 June 2023) was published showing an initial decrease in bacterial pathogens after one month of therapy, but persistence of infection at 6 months [89]. Encouragingly, a small single-center pilot study of 58 pwCF starting ETI found a reduction in the proportion of sputum cultures positive for *Aspergillus* sp. and a reduction in total IgE and anti-*Aspergillus* precipitating antibodies, although specific IgG and IgE antibodies did not decrease [90]. Ongoing longitudinal studies will provide important additional information, although the reduction in sputum availability must be considered in designing and interpreting all of these studies.

## 10. Summary

*Aspergillus fumigatus* presents a clinical challenge in pwCF, with some people seemingly unaffected while others potentially experiencing substantial morbidity from ABPA or fungal bronchitis. Future studies are needed to clearly identify people at risk of disease progression associated with Af, clearly define the characteristics of Af disease in CF, and to improve the treatment regimens directed towards *Aspergillus* and other fungi. CFTR modulators, particularly ETI, have changed the landscape of CF. While the prevalence of infection may decrease, the risk of acquisition and persistence of infections are likely to remain elevated for many people with CF and given the advanced eukaryotic nature of Af and the potential for an allergic host response, there is concern that this organism will remain a difficult-to-treat pathogen. The use of ETI has also made detecting fungi more difficult given a reduction in the availability of lower airway sputum samples from pwCF. In addition, treatment with antifungals is complicated by a significant drug–drug interaction between ETI and azole therapy, making treatment decisions difficult and clinically complicated. Finally, some people with CF are not able to take modulators due to unresponsive CFTR-variants or drug intolerance and will remain at an elevated risk of ABPA and persistent Af infection. Given these findings, the Af disease in pwCF needs further understanding at all levels and tiers of investigation to better understand the mechanisms of this organism in the CF airway, the prevalence and impact of this organism on CF health, and treatment guidelines in the ETI era.

## Data Availability

No new data were created or analyzed in this study. Data sharing is not applicable to this article.

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
