# Peer review of "Infection, Allergy, and Inflammation: The Role of Aspergillus fumigatus in Cystic Fibrosis"

_microorganisms, 2023, doi:10.3390/microorganisms11082013_

Round 1
Reviewer 1 Report
This review deals with the impact of Aspergillus spp colonization of airways in CF patients. Although a lot of information and investigation is lacking currently on this topic, the review is interesting, well written and easy to read. I have not concerns on the text for publication
Author Response
Thank you very much for your review and positive comments.
Reviewer 2 Report
This review is a comprehensive, detailed, well-structured and well-written summary covering the vast majority of aspects related to Aspergillus-associated disease in CF.
After minor revision, it can give the reader a complete overview of the field and may thus be a valuable source of literature even for readers who are not very familiar with the theme.
1. In chapter 7, inasive aspergillosis/A.f. pneumonia in pwCF without immunosuppression as a relatively rare but possible clinical entity should be mentioned.
2. Also in chapter 7, aspergilloma shoud be mentioned as a A.f.-related clinical entity.
3. Chapter 6 should mention the CFTR-related impairment of chlorination in neutrophils with resulting dysregulation of neutrophils.
4. Also in chapter 6, CFTR-dependent impairment of T and B cell reulation are should be mentioned as part of dysregulated host response.
5. In chapter 4, dehydration of the airway surface layer (ASL) should be added.
Line 89: ....characterized by an allergic..., ...hypersensitivity, and Ig-E....
Line 236: ...rarely seen in...
Author Response
Response to Reviewer 2 comments
Point 1: This review is a comprehensive, detailed, well-structured and well-written summary covering the vast majority of aspects related to Aspergillus-associated disease in CF. After minor revision, it can give the reader a complete overview of the field and may thus be a valuable source of literature even for readers who are not very familiar with the theme.
Response 1: We appreciate the reviewer’s comments
Point 2: In chapter 7, invasive aspergillosis/A.f. pneumonia in pwCF without immunosuppression as a relatively rare but possible clinical entity should be mentioned.
Response 2: We have added the following to line 244:
“…although invasive aspergillosis pneumonia and aspergillomas (fungal balls) have also been reported in immunocompetent individuals with ABPA.”
Point 3: Also in chapter 7, aspergilloma should be mentioned as an A.f.-related clinical entity.
Response 3: We have added the following to line 306:
“Invasive pulmonary aspergillosis, chronic necrotizing aspergillosis, and aspergillomas caused by invasion of lung parenchymal tissue by Aspergillus are most often found in individuals with severe immunocompromise. However, individuals with underlying chronic lung disease, including chronic obstructive pulmonary disease (COPD) and CF, are also at risk, likely due to lung structural injury (e.g., cavitary disease, bronchiectasis). Antifungals are the primary treatment for invasive pulmonary aspergillosis. In the case of aspergillomas, surgical resection is often required.20”
Point 4: Chapter 6 should mention the CFTR-related impairment of chlorination in neutrophils with resulting dysregulation of neutrophils.
Point 5: Also in chapter 6, CFTR-dependent impairment of T and B cell regulation are should be mentioned as part of dysregulated host response.
Response to point 4 and 5: We have added the following to line 200 (new statements underlined):
“CFTR is also present in neutrophils, lymphocytes, and macrophages. Dysfunction of CFTR likely directly impairs host inflammatory response through abnormalities in chlorination, altered inflammatory pathways, and impaired pathogen killing.25 This leads to a vicious cycle of bacterial growth in the abnormal mucus, increased cell debris and contents from increased neutrophil activity, utilization of cell materials by the bacteria, proliferation of bacteria, and then further recruitment of neutrophils.1, 60 CFTR-deficient T-cells and B-cells also may favor a Th-2 and exaggerated IgE response predisposing individuals to ABPA.25
Point 6: In chapter 4, dehydration of the airway surface layer (ASL) should be added.
Response 6: We have now added to following at line 108:
“CFTR dysfunction with impaired chloride and bicarbonate transport also results in dehydration of the airway surface liquid layer impairing cilia movement. These changes restrict mucociliary clearance and allow persistence of antigenic agents.”
Minor comments:
Line 89: ....characterized by an allergic..., ...hypersensitivity, and Ig-E....
Line 236: ...rarely seen in...
Response to minor comments: These changes were made to the manuscript
